# The Advances in Phospholipids-Based Phase Separation Gels for the Sustained Release of Peptides, Proteins, and Chemotherapeutics

**DOI:** 10.3390/pharmaceutics16070875

**Published:** 2024-06-29

**Authors:** Jianxia Dong, Xueru Zhou, Qing Li, Ruohui Zheng, Jing Chen, Yuzhe Liu, Xin Tong, Zhuoya Wan, Tao Gong

**Affiliations:** 1Department of Pharmacy, West China Hospital, Sichuan University, Chengdu 610041, China; 15802891207@163.com (J.D.); 18381476874@163.com (Q.L.); 2West China School of Pharmacy, Sichuan University, Chengdu 610041, China; hersly@163.com; 3Department of Pharmaceutical Sciences, School of Pharmacy, University of Pittsburgh, Pittsburgh, PA 15261, USA; ruz33@pitt.edu (R.Z.); jic110@pitt.edu (J.C.); xit25@pitt.edu (X.T.); 4Department of Mechanical Engineering and Materials Science, University of Pittsburgh, Pittsburgh, PA 15261, USA; yul155@pitt.edu

**Keywords:** phospholipids-based phase separation gels, sustained release drug delivery, in situ forming implants, peptides and proteins, chemotherapeutics

## Abstract

Implantable drug delivery systems formed upon injection offer a host of advantages, including localized drug administration, sustained release, minimized side effects, and enhanced patient compliance. Among the various techniques utilized for the development of in situ forming drug implants, solvent-induced phase inversion emerges as a particularly promising approach. However, synthetic polymer-based implants have been associated with undesirable effects arising from polymer degradation. In response to this challenge, a novel category of drug delivery systems, known as phospholipids-based phase separation gels (PPSGs), has emerged. These gels, characterized by their low initial viscosity, exhibit injectability and undergo rapid transformation into in situ implants when exposed to an aqueous environment. A typical PPSG formulation comprises biodegradable components, such as phospholipids, pharmaceutical oil, and a minimal amount of ethanol. The minimized organic solvents in the composition show good biocompatibility. And the relatively simple composition holds promise for industrial-scale manufacturing. This comprehensive review provides an overview of the principles and advancements in PPSG systems, with specific emphasis on their suitability as drug delivery systems for a wide range of active pharmaceutical ingredients (APIs), spanning from small molecules to peptides and proteins. Additionally, we explore the critical parameters and underlying principles governing the formulation of PPSG-based drug delivery strategies, offering valuable insights on optimization strategies.

## 1. Introduction

Controlled release delivery strategies are meticulously engineered to precisely modulate the release of drugs at an ideal rate, ensuring the maintenance of drug efficacy over a defined and extended period [1]. In situ forming (ISF) drug delivery implants have emerged as a prominent and highly frequent controlled release delivery system in recent years. These ISF systems offer a multitude of therapeutic advantages, including site-specific action, sustained release profiles, and straightforward preparation processes. Consequently, they contribute to reduced dosing frequency, enhanced patient compliance, fortified and prolonged antibody immune responses, and a preference for immunological memory-based responses [2,3].

There are many kinds of ISF delivery systems under development. In general, these systems are made from viscous materials that solidify into a semisolid depot upon dosing and, therefore, can control the delivery profiles of the incorporated drugs to achieve long-term therapeutic action [2,3]. Traditional ISF systems are composed of water-insoluble biodegradable polymers [4,5]. Commonly used ISF biodegradable polymers include polyhydroxy acid, polyanhydride, polyorthoester, and their derivatives. Among them, poly(L-lactide) (PLA), poly(epsilon-caprolactone) (PCL), and poly (lactide-co-glycolide) (PLGA) polymers are widely used in ISF systems [4,5]. These water-insoluble polymers are first dissolved in drug-incorporated, organic, water-miscible, and biocompatible solvents. Once this polymer system is introduced into an aqueous environment, such as in vivo injection, the organic solvent will dissipate out of the system, while water infuses into the system via diffusion [6]. This exchange of solvents results in solution-to-gel transformation and implant formation by causing polymer precipitation, which in turn controls the drug release rate. However, there are still many challenges to overcome in terms of polymer-based ISF systems, such as the rather complicated manufacturing processes involved, expensive raw materials that increase the final product cost, and reported local inflammation due to their metabolites [7,8,9,10]. Apart from polymer-based ISF systems, non-polymeric biodegradable systems have also been reported as ISF systems, such as the sucrose acetate isobutyrate (SAIB)-based [11,12] ISF system. Compared to polymer-based ISF systems, the SAIB-based ISF system utilizes less expensive raw materials and has more straightforward manufacturing processes. In polymer-based ISF systems, upon administration, the solvent diffuses out leaving an adhesive and viscous matrix. However, some other issues have been raised in regard to this system, such as the initial burst release profile due to the lag period between the injection and formation of the implant. The burst release profile was only lowered with the inclusion of PLGA polymers [12].

Another classification of ISF material is hydrogel, noted by its high water concentration in the gel system due to crosslinked hydrophilic polymers [13]. Despite having good biocompatibility and various mechanical properties, hydrogels are most useful when the APIs are water soluble [14]. Notably, achieving sustained release in relation to hydrophilic drugs poses a challenge due to the aqueous nature of the physiological environment within the human body. Moreover, both hydrogels and polymer-based ISF gel systems use diverse crosslinking agents and materials, meaning the majority of the excipients are not FDA approved and they necessitate additional assessments and approvals [15].

Therefore, to address the abovementioned issues, it is necessary to develop novel ISF strategies that: (1) utilize raw materials with a lower cost, (2) retain biodegradable properties, (3) improve the safety profile of the final products, and (4) ease the scale-up for mass production. In addition, there are other concerns related to traditional ISF systems. ISF delivery systems use solvents that can efficiently dissolve the polymer and be miscible with aqueous water, as well as body fluids. However, these organic solvents also irritate the injection site. Thus, investigating novel solvents to develop ISF systems with less irritation is another research direction to improve the final products, where various factors, including the solubility of the raw materials, toxicity, and biocompatibility, and the ability to maintain system stability should be considered.

In the last few decades, researchers have developed a novel category of drug delivery systems, PPSGs, for the controlled release of peptides, proteins, and chemotherapeutics (Table 1). When administrated in liquid form, the PPSG system undergoes phase separation, resulting in the formation of a solid or semisolid gel at the injection site due to the in situ water–ethanol exchange between the surrounding body tissues and the PPSG system (Figure 1). Mechanistically, the solution-to-gel transition exhibits several benefits in comparison to other gel formation mechanisms [16]. The dissolved phospholipid molecules self-assemble into a three-dimensional network to form a gel-like structure. On one hand, the rigorous physical and chemical conditions, such as extreme temperature, pH, and ionic strength, commonly employed in other in situ gel formation methods, can be avoided with this gel formation method [17,18,19]. On the other hand, the components of gel formation are commercially available and obtained easily. PPSG formation excipients include biodegradable lecithin/physiological lipids that are widely used in FDA-approved products, with low toxicity and excellent biocompatibility for clinical use [20]. We propose that this PPSG system can solve previous issues and represents great potential as being a next-generation ISF delivery strategy. The main objective of this review is to summarize the study outcomes for recently developed PPSGs, along with the wide range of biomedical applications.

## 2. Mechanism of PPSG Formation

PPSG is a special ISF gel (Figure 2a), which is composed of three basic components: phospholipids, low-percentage ethanol, and pharmaceutical oil. Phospholipids are the major components of cell membranes and serve as a drug depot in PPSGs. Ethanol works as a solvent for phospholipids to increase the spreadability (reduce viscosity) and, therefore, ensure a smooth injection. When this system comes into contact with an aqueous environment, a solvent exchange occurs immediately at the interface, consequently inducing solidification of the PPSG (Figure 2b). A layer of the shell structure quickly forms at the surface of the gel. Then, the internal ethanol dissipates out of the system, while water ingresses gradually via diffusion. This exchange of solvents results in an increase in viscosity and solution-to-gel transition, which turns the phospholipids into a drug depot. (Figure 1). The ternary phase diagram of PPSG shows a big phase transition area (Figure 2c), which also illustrates that the PPSG can easily transit to a gel in an aqueous environment. Furthermore, when a PPSG is observed using atomic force microscopy (AFM), there are no bright spots in the AFM graph, indicating that the phospholipids in the PPSG are uniformly dissolved and well distributed. After PPSG transforming into a gel occurs when it comes into contact with an aqueous environment, the AFM graph shows many bright spots, which illustrates the presence of solidified phospholipid particles of a certain size. This result suggests that the transformation of PPSG into a gel leads to the precipitation and solidification of the phospholipids, consequently becoming a reservoir for the sustained release of drugs (Figure 2d). However, a local application of ethanol above 50% v/v may be irritating to the skin [42]. Thus, Zhang T et al. evaluated the irritation of PPSG in rabbits, which showed that the presence of a minimized amount of ethanol might cause some, but an acceptable level of, irritation in rabbits [25].

Pharmaceutical oil is another component of PPSG, which aims to decrease the viscosity and reduce the amount of ethanol required [25]. As a result, the viscosity of the initial PPSG system could be decreased for easier injection. But it also delays the speed of the phase transition, because it can slow down the water–ethanol exchange process. Meanwhile, adding pharmaceutical oil also significantly reduces the total usage of ethanol. In this way, it reduces skin irritation due to ethanol [25]. Upon exposure to the aqueous solution, the ethanol in PPSGs will dissipate out of the system, while body fluids from the outside make their way into it via diffusion, which leads to phospholipid precipitation and the formation of a drug depot.

Viscosity, solvent diffusion, and the micro-structure are the basis of low initial burst release and long-term release properties. Phospholipids (PLs) can be easily dissolved in ethanol at a concentration range from 50% to ~80% (g/g), and a positive correlation exists between the concentration of PLs and gel viscosity. Usually, the higher the concentration of the PLs helps to avoid burst release and contributes to the controlled release of drugs from the PPSG during the fast phase transition.

The solvent diffusion speed is a key factor in the phase transition and drug release of PPSGs. According to the 3D phase diagram published in our previous research, PPSGs have a very large phase transition area (68.89%), indicating their exceptional capacity for gel formation upon exposure to an aqueous solution (Figure 3) [25]. The onset of PPSG formation is around 120 min in vitro [25]. PLs and pharmaceutical oil at a high concentration prevent the aqueous solution from entering the inner core of the gel, which would reduce the dissolution rate of the encapsulated drugs. In addition, the viscosity of this system is less than 200 cp (comparable to clinically accepted injections) and maintains good liquidity [43].

## 3. The Excipients of PPSG Systems

The excipients of PPSG systems are summarized in Table 2. Phospholipids used in PPSGs are mostly phosphatidylcholines (PCs). Pharmaceutical phosphatidylcholine is often derived from natural sources, like soybeans or egg yolks, and can undergo purification processes to meet pharmaceutical standards. Its chemical structure consists of a glycerol backbone, to which two unsaturated fatty acid chains are esterified and a phosphocholine head group is linked to the glycerol’s third hydroxyl group. Phospholipids are the key phase-change materials in PPSGs.

The pharmaceutical oils used are essential. They are responsible for lowering the viscosity, while maintaining the sustained release. For the oil itself, both the hydroxy substitution degree and the fatty acid chain itself (chain length and saturated or not) affect the drug crystallinity and release profile. In the PPSG system, medium-chain oil, soybean oil, oleic acid esters, and sorbitan monooleate could be used as the oil phase. Each of these oils is unique in terms of their own properties and needs to be carefully examined for use in different systems.

Medium-chain triglycerides (MCT) are a category of lipids characterized by the presence of three saturated carbon chains attached to a glycerol backbone, with the carbon chains having a length ranging from six to twelve carbons. The most common MCTs include caproic acid (C6), caprylic acid (C8), capric acid (C10), and lauric acid (C12).

Through the process of esterification, oleic acid, an unsaturated fatty acid present in various vegetable oils, becomes linked to glycerol. This linkage results in different substitution degrees, giving rise to oils with distinct amphiphilic characteristics, specifically glycerol monooleate (GMO), glycerol dioleate (GDO), and glycerol trioleate (GTO). In a head-to-head comparison, it was found that the different amphiphilic properties lead to gel formations with different properties [27]. GTO gel had a lower initial release rate and a more stable release profile, a slower solvent diffusion speed, and less skin irritation [27].

## 4. PPSGs as Peptide and Protein Delivery Systems

Protein and peptide drugs are widely employed in the treatment of cancer, cardiovascular diseases, and diabetes, as well as many other diseases [44]. With advantages such as lower clinical doses, higher pharmacological activities, and fewer side effects, druggable proteins and peptides are considered to be promising research targets for developing next-generation therapeutic agents [44]. However, there are several challenges in the development of protein and peptide drugs [26,45,46]: (1) administration is limited to injections due to their digestible nature, requiring higher patient compliance; (2) stability varies significantly under different pH environments and the appearance of endogenous factors, depending on the chemical structure; and (3) the dosing frequency is high because of their relatively short half-life. As a result, developing injectable peptide and protein formulations with extended release profiles and long-term therapeutic effects has been a topic of broad interest among the pharmaceutical community. To overcome the barriers mentioned above, many sustained release formulations have been developed in recent years, such as Eliaard^®^ (leuprolide acetate) (Camurus AB, Lund, Sweden), Somatuline^®^ Depot (lanreotide) (Ipsen, Boulogne-Billancourt, France), and Sandostatin LAR^®^ (octreotide) (Novartis, Basel, Switzerland). However, the preparation of polymeric nanoparticles and microspheres is rather complicated, not to speak of the elevated price due to this, while their metabolites are known to cause local sterile inflammation [7,8,9]. Thus, there is a need to develop novel sustained release peptide/protein-based delivery strategies. In this section, we summarize the PPSG-based peptide and protein delivery systems, which are also summarized in Table 3.

### 4.1. Delivery of Octreotide Acetate (OCT)

Octreotide is an analog of natural somatostatin, which has demonstrated enhanced potency, as well as being metabolically stable, than its parent drugs [47]. OCT has been extensively used in the treatment of cancer and growth hormone-related diseases. As a therapeutic agent for acromegaly, carcinoid syndrome, and endocrine tumors of the GI tract, OCT has demonstrated its capability to inhibit the secretion of insulin, glucagon, and gland/growth hormones. However, OCT has an in vivo half-life of less than 2 h, and its therapeutic effect drops rapidly following subcutaneous injection or the cessation of infusion [48]. To improve the therapeutic performance of OCT, long-acting OCT formulations, namely Sandostatin^®^ LAR [49] (Novartis, Basel, Switzerland) and Lanreotide Autogel^®^ [50] (Ipsen, Paris, France) are currently available on the market. Sandostatin^®^ LAR is a once monthly depot formulation based on PLGA microspheres, while Lanreotide Autogel^®^ is a supersaturated aqueous gel based on polypropylene. Furthermore, an octreotide ISF depot (CAM2029) developed by Novartis and Camurus (Lund, Sweden), based on FluidCrystal^®^ technology (Camurus AB, Lund, Sweden), has achieved positive phase 3 data in patients with acromegaly [51,52].

A readily injectable, biocompatible, and efficient ISF gel platform material for the sustained delivery of OCT has been developed [25,26,27]. Experimental results have shown that OCT was extremely soluble in ethanol and in the gel systems. The appearance and viscosity of the gel barely changed after OCT had been added. As an alternative sustained release preparation type of octreotide microsphere, octreotide loaded in situ with gel (OCT-gel) has great research prospects due to its huge technological and economic advantages [53,54,55]. Regarding the PPSG system, 85% ethanol in PBS was used as the solvent. Zhang T et al. evaluated the irritation of PPSG in rabbits, which showed that the presence of ethanol might cause some, but an acceptable amount of, irritation in rabbits [25]. OCT was dispersed homogeneously in the PPSG pre-gel solution to afford OCT-loaded PPSG (OCT–PPSG) after a single subcutaneous injection, which displayed controlled and sustained release profiles for up to 30 days in rats, rabbits, and Beagle dogs. Another major advantage of PPSG is its sustained release profile without a remarkable initial burst effect. OCT–PPSG showed a less significant burst phase, followed by a steady plasma concentration of OCT compared with Sandostatin^®^ (LAR) in Beagle dogs. The release of OCT–PPSG occurred in a sustained and controlled manner, and the cumulative release of OCT from OCT–PPSG within 72 h was about 10% in a PBS-only medium [26]. To evaluate the therapeutic efficacy of OCT–PPSG, we selected hepatocellular carcinoma (HCC) as the disease model, which is the fifth most common cancer in the world and the third most common cause of cancer-related death [56]. OCT–PPSG showed remarkable antitumor efficacy in both a primary rat model and a xenograft mouse model of HCC [26]. PPSG thus represents a promising and viable ISF gel platform material for the long-term sustained release of peptides and protein drugs.

Exenatide (synthetic exendin-4, EXT) is a peptide that has been proven to be an efficient and safe GLP-1 receptor agonist for the treatment of type 2 diabetes mellitus [57,58]. The EXT solution for injection (Byetta^®^, Amylin, CA, USA) has been marketed as an antidiabetic drug upon its approval from the Food and Drug Administration (FDA) in 2005 and the European Medicines Agency (EMA) in 2006. However, the dosing frequency is not patient friendly because Byetta^®^ requires twice-daily injections, which limits the use of this peptide. As various GLP-1 receptor agonists crowd the marketplace, it has been revealed in real-world studies that extended dosing intervals improve patient adherence [59]. Various half-life extension strategies have been employed, among which, microspheres and implant-based formulations are frequently explored [18,60,61]. An EXT-loaded microsphere has been marketed as Amylin (Bydureon^®^) for the treatment of type 2 diabetes. It significantly reduces the dosing frequency, as a first-time reported once-weekly long-acting injectable formulation [61]. On the other hand, once-yearly titanium implants (ITCA 650) have been repeatedly rejected by the FDA due to acute kidney injury and cardiovascular risks [62]. Hence, it is still of great value to explore safe and biocompatible delivery systems designed for GLP-1 receptor agonists.

Among various formulations, ISF gels have demonstrated several advantages when it comes to delivering antidiabetic peptides. These advantages include the ease of preparation and administration, reduced dosing frequency, and improved patient compliance. A noteworthy development in this field is the creation of a once-weekly injectable synthetic co-polymer thermo-gel (PLGA–PEG–PLGA) by Yu et al. in 2013. This gel has been reported as a long-acting formulation, capable of maintaining its therapeutic effect for seven consecutive days following a single subcutaneous injection [63]. Additionally, Hu et al. reported that PPSG loaded with an antidiabetic peptide (EXT) exhibited long-term glucose control. PPSG effectively maintained blood glucose levels for 15 days in chemically induced diabetic rats and 21 days in db/db mice, using a spontaneous rodent model for type 2 diabetes [21]. Notably, PPSG/EXT achieved this without any burst release. A single subcutaneous injection of PPSG/EXT demonstrated a hypoglycemic effect comparable to twice-daily injections of EXT solution in the diabetic rat model. This not only ensures effective treatment, but also significantly improves patient compliance. These advancements in ISF gels offer promising solutions for long-term antidiabetic peptide delivery, providing convenience and efficacy for patients.

### 4.2. PPSGs for the Long-Term Delivery of Other Peptides

PPSGs have demonstrated remarkable potential for the sustained delivery of various proteins, including leuprolide acetate (LA) [22,23], thymopentin (TP5) [24], and insulin [29], offering long-lasting therapeutic effects and excellent biocompatibility. LA-loaded PPSG [22] showed sustained release for 30 days and a significantly reduced initial burst with decent suppression of testosterone in rats and dogs, compared with a commercial depot formulation of leuprolide (Figure 4). TP5-loaded PPSG demonstrated a month-long drug release profile, both in vitro and in vivo, underscoring its sustained and controlled release properties [24]. A single dose of TP5 PPSG (15 mg/kg, s.c.) achieved an immunoregulatory effect comparable to that of repeated TP5 solution injections (0.6 mg/kg per day, s.c.) over 14 consecutive days. PPSG has also been applied to insulin delivery, resulting in a sustained release profile and long-lasting hypoglycemic effects in diabetic rats [29]. This outcome holds significant potential for enhancing patient compliance and reducing hospitalization costs for individuals with type 1 and type 2 diabetes mellitus (DM), who typically require daily insulin injections to maintain stable blood glucose levels [64]. These achievements underscore the versatility and promise of PPSG as a delivery platform for a range of therapeutic proteins, facilitating prolonged and effective treatment.

### 4.3. PPSGs for the Delivery of Vaccines

Vaccination is one of the breakthrough achievements in modern medicine, saving the lives of more than 3 million people per year [65]. The global pandemic period highlighted the urgent need for effective and safe vaccines. Moreover, the co-administration of protein antigens and adjuvants can remarkably increase immunogenicity, which has spurred the rapid development of numerous antigen delivery systems. Among the reported formulations [66,67,68,69,70,71], incomplete Freund’s adjuvant (IFA) has been used for more than 60 years as a water-in-oil emulsion for co-delivering antigens and pathogen-associated molecular pattern molecules [72]. Despite the strong, long-lasting IgG responses stimulated, IFA has not been approved due to its potential severe side effects, which include inflammation, necrosis, local irradiation, persistent painful granulomas, sterile abscesses, and cysts at the injection sites [73,74]. This points out the direction of the following works, and it is highly demanded that an antigen–adjuvant delivery system with an IFA-like high potency, along with much lower toxicity, is developed [28].

It has been demonstrated that PPSG loaded with model antigen ovalbumin (OVA) supported sustained OVA release in mice, which can last nearly one month (Figure 5) [28]. Immunizing mice with a single injection, PPSG transformed in situ into a drug depot for controlling the release profile of the loaded vaccines, which then elicited robust and persistent antigen-specific humoral immune responses and strong immunogenic memory. More specifically, PPSG/OVA-induced a strong and persistent increase in titers of OVA-specific IgG, IgG1, and IgG2a. To improve the efficacy of OVA alone, the co-delivery of CpG oligodeoxynucleotide (CpG-ODN or CpG) as an adjuvant acted in synergy with PPSG/OVA to elicit a faster and stronger immune response, which was supported by the significantly enhanced antibody titers, more recruited dendritic cells (DCs) in the injected sites, and more activated DCs in the draining lymph node. In addition, this work has also demonstrated that the immunization of PPSG/OVA/CpG can lead to a high frequency of memory T cells and a potent memory immune response. In terms of its safety profile, PPSC/OVA/CpG caused much less irritation at the injection site in comparison to IFN/OVA/CpG. With a high dosage of CpG (50 μg), PPSG/OVA/CpG showed no systemic toxicity such as in regard to the lymph nodes, spleen, or liver, nor did it affect body weight. In contrast, IFA/OVA/CpG induced severe lymphadenopathy, liver inflammation, and significant splenomegaly. In summary, the PPSG/OVA/CpG-based delivery system appears to be a more effective and safer vaccination for delivering antigens and adjuvants.

## 5. PPSGs as Chemotherapy-Based Delivery Systems

Sustained release systems offer significant advantages by reducing the frequency of drug administrations, particularly in the treatment of cancer and chronic diseases, like nonalcoholic fatty liver disease. This is especially critical for patients who may have difficulties adapting to frequent dosing regimens, including children and individuals with mental disorders. PPSGs emerge as a promising platform for the development of innovative sustained release chemotherapy-based delivery strategies. In this section, we present a summary of the novel chemotherapy approaches leveraging PPSG delivery systems, highlighting a diverse range of potential applications.

### 5.1. Application of PPSGs for the Prolonged Delivery of Anticancer Drugs

Intratumoral injectable ISF gel presents a highly promising approach for regional chemotherapy delivery, addressing several common issues associated with anticancer drugs. Some of these issues are detailed as follows. (1) Accumulating drugs at the desired site: This approach allows for the targeted accumulation of drugs at specific sites within the body, while simultaneously minimizing their systemic distribution [16,75,76]. (2) Enhancing drug concentration in tumor sites: By focusing drug delivery directly to the tumor site, this method significantly increases drug concentrations within tumors. Moreover, it mitigates dose-dependent toxicities, as drug distribution to normal organs and tissues is reduced compared to systemic administration [3]. (3) ISF degradable networks: The concept of ISF degradable networks holds promise not only in tissue engineering, but also in drug delivery applications. To achieve site specificity, prolonged action, and improved patient compliance with PPSGs, our research group has developed a PPSG formulation for the sustained delivery of anticancer drugs. This innovative approach represents a significant step toward improving the effectiveness and safety of cancer treatment.

Wu et al. developed a Dox-loaded phospholipid-based phase separation gel (Dox PPSG, referred to as PME-D), using a mixture of E80, MCT, and ethanol in a mass ratio of 72:17:11 (Figure 6) [16]. This gel formulation served as a drug reservoir within the tumor site, effectively controlling the release of Dox for a duration exceeding 14 days following a single injection. Notably, PME-D demonstrated remarkable tumor growth inhibition throughout the 14-day treatment period, all achieved without any burst release.

Chen et al. also developed a paclitaxel-loaded phospholipid-based gel for the local treatment of glioma [30]. They improved the PPSG system with several strategies, including: (1) using less solid phospholipids and more oil to lower the viscosity, (2) less ethanol to achieve better tolerability, and (3) introducing triacetin (GTA) as not only a chemotherapeutic adjuvant agent but also a viscosity reducer [30,77]. The PTX release profile was close to TMZ [78] and gemcitabine [79], with a relatively faster initial release and a sustained low-dose release in the later stage. The PTX release was sustained for 15 days in vivo, achieved a prolonged survival time, and subsequently alleviated the side effects. The concentration of ethanol was reduced to 7% as well, to avoid irritation [30]. Yang et al. further improved the PPSG system by delivering magnesium oxide (MgO) and 5-fluorouracil (5-FU) to target the tumor microenvironment and tumor cells, while decreasing the ethanol concentration to 7% to reduce irritation [31]. This system could successfully prolong the release of 5-FU and MgO for antitumor and anti-metastasis, by killing the tumor and acidic tumor microenvironment neutralization [31,80].

Wu et al. developed a PPSG system co-loaded with the antitumor agent pirarubicin (THP) and the cyclooxygenase-2 (COX-2) inhibitor celecoxib (CXB), which inhibits the postoperative recurrence and metastasis of breast cancer (Figure 7) [81]. After injection, the gel underwent a spontaneous phase transition and formed a drug reservoir that fitted the irregular surgical incisions perfectly. In vivo, the drug release was observed to last as long as 25 days for both THP and CXB. The in vivo therapeutic efficacy was evaluated in 4T1-bearing BALB/c mice. THP–CXB–IPG showed considerable inhibition of residual tumor growth after surgery and reduced the incidence of pulmonary metastasis. Moreover, it reduced the systemic toxicity of the chemotherapeutic agents.

### 5.2. PPSG Extends the Duration of Action for Other Low Molecular Weight Compounds

PPSGs have proven valuable in enhancing the safety and extending the in vivo retention of 2,4-dinitrophenol (DNP), a mitochondrial uncoupler used against nonalcoholic fatty liver disease (NAFLD) [37]. This formulation, PPSG–DNP (DNP–LC gel), effectively prevents burst release, achieving a Cmax value of 14.84 µg/mL post-administration and sustaining release for 12 h. By mitigating the risk of frequent hyperthermia, DNP–LC gel can contribute to lower mortality rates when DNP is applied clinically.

Sustained release of local anesthetics at surgical sites following a single injection is a favored approach for controlling post-surgical pain [82]. Li et al. developed a PPSG system (RO–PPTG) capable of delivering the amide anesthetic ropivacaine (RO) to surgical sites for prolonged postoperative analgesia [34]. RO–PPTG exhibits sustained release with minimal initial burst release, achieved through the rapid in situ formation of highly viscous PPTG gel. In pre-clinical studies, RO–PPTG showed tactile sensation recovery commencing at approximately 6 h, while maintaining anesthesia for more than 36 h, a duration 3.6-fold longer than traditional unformulated RO solutions.

Widespread availability and patient adherence to effective malaria treatment remain significant challenges in developing countries. Sustained release delivery systems for malaria preventative drugs hold promise as a potential solution [83], with research efforts spanning two decades [84,85,86]. Notably, an orally administrated capsule encapsulating ivermectin, designed to last over 14 days within the gastrointestinal tract, was reported by Robert Langer’s group [87]. This capsule significantly enhanced the efficacy of mass drug administration (MDA) and underscores the potential of sustained release systems to revolutionize malaria treatment options.

Luo et al. [38] introduced an injectable meshy gel system designed to extend the release duration of ivermectin to at least 30 days. The modification of chondroitin sulfate with octadecylamine (C18) served to restrain the phospholipids, eliminating initial burst release. The meshy gel system, gelatinized through ethanol diffusion upon aqueous exposure, led to the precipitation of water-insoluble materials (phospholipid, CS-C18, and MCT) within 10 min, forming a solid drug depot. This long-acting release meshy gel system holds promise as an effective tool for protracted warfare against malaria (Figure 8).

## 6. Conclusions

ISF implants have gained popularity due to their therapeutic advantages, including site-specific action, sustained release, and improved patient compliance. Traditional ISF systems often utilize biodegradable polymers such as PLGA. However, they are plagued by issues like burst release and the generation of unwanted metabolites. To address these challenges, a next-generation ISF delivery strategy is proposed in the form of PPSGs.

PPSGs, initially in liquid form, undergo a water–ethanol exchange upon administration, resulting in the formation of solid or semisolid phosphocholine-based gels at the injection site. Studies have demonstrated the success of PPSG systems as versatile delivery platforms, particularly for peptides and proteins, showcasing their potential applications in chronic metabolic disorders. Furthermore, PPSGs exhibit versatility as carriers for antigens and various chemical entities, enabling treatments ranging from metabolic disorders and cancer to autoimmune diseases and mental disorders.

Both in vivo and in vitro experiments have consistently shown the PPSG system’s ability to extend drug release over weeks or even months. The systematic design of excipients and the precise control of crystallinity within the PPSG system offers a promising solution to challenges encountered in traditional delivery systems. Importantly, by relying on FDA-approved excipients, PPSGs hold the potential to streamline the development of commercial products that could provide substantial benefits to patients on a global scale.

## 7. Challenges and Perspective

PPSGs share essential parameters with conventional ISF systems, while also introducing distinctive challenges. In contrast to extensively studied polymer-based gels, PPSG systems present a range of parameters that necessitate further investigation, including the kinetics of phase formation and the characterization of crystalline forms.

### 7.1. Burst Release

The concern regarding initial burst release in ISF depots is indeed a significant challenge, especially when considering synthetic polymer-based systems [2]. This burst release phenomenon goes against the principles of sustained drug release, and it can have critical implications for both the therapeutic effectiveness and economic aspects of the treatment, particularly when dealing with drugs that have a narrow therapeutic window. The issue becomes even more pressing because the drugs released during the burst phase are essentially wasted, which can lead to an increased economic burden, especially when the APIs hold substantial value.

While there is no definitive explanation for burst release in monolithic polymeric systems, it is widely accepted that in ISF systems, the inability of the gel forming immediately after injection is a leading cause of this initial burst release. This means that the drugs are not completely encapsulated before they are exposed to the local physiological environment [88,89]. In designing a successful ISF system, the gelation process plays a pivotal role as it determines the morphology and release behavior of the drug. However, the hydrophobic nature of oils and lipids can result in a slower gelation process. Therefore, finding innovative solutions to address the delayed solidification time is essential.

In the reports discussed within this review, a phenomenon known as ‘burst release’ was observed in lipid-based systems. This suggests that, despite being free from toxic metabolites resulting from polymer degradation, lipid-based systems may not entirely resolve the issue at hand. Drawing from the experiences and strategies employed in both polymer-based gels and liquid crystal systems, several potential avenues for improvement can be explored. These include combining lipids and polymers [12,90], optimizing solvents to enhance the diffusion capacity [27,36], or adding surfactants [41]. Furthermore, an alternative approach worth noting is the use of FluidCrystal^®^ technology in the development of Octreotide subcutaneous depots, which has been reported to exhibit initial rapid release kinetics instead of the typical burst release observed in human trials [52].

### 7.2. Nanostructure

To ensure the precise control of critical process parameters and the preservation of a stable drug release profile, it is imperative to undertake a comprehensive characterization of the specific nanostructure of PPSG. For example, a synchrotron small-angle X-ray diffuser and a transmission electron microscope could be used to analyze the nanostructures. The release profile of encapsulated cargoes is predominantly influenced by the mesophase geometry or dimensions within the gel matrix. Research studies have consistently demonstrated that the hexagonal phase, in contrast to the cubic phase, exhibits a notably slower drug release rate [91]. It is essential to acknowledge that any alteration of the excipients will not only impact the gelation process, but also influence the matrix morphology, diffusion coefficients, and the overall drug release profile. Therefore, when formulating a specific drug, careful consideration should be given to both the core components and additives. Furthermore, phospholipids, often derived from natural sources, can exhibit heterogeneity and exhibit significant batch-to-batch variations, resulting in varying characteristics at the molecular level. Consequently, it is imperative to exercise added caution in ensuring material consistency.

### 7.3. Patient Compliance and Clinic Usage

A significant limitation of polymer-based ISF gels lies in their inherent viscosity, often necessitating the expertise of a specially trained nurse to pre-emulsify or preheat the formulation to 37 °C before injection [92]. The next generation of ISF gels is anticipated to address this issue, aiming to simplify their application for improved patient compliance and economic benefits. Additionally, it is crucial to acknowledge that the drug release profile is profoundly influenced by the configuration of the drug depot, contingent on both the wound’s shape and the rate at which the syringe is administered. Parameters such as drug loading and injection volume assume critical roles for different types of incisions.

Fluid exchange and gel formation rates exhibit variations across diverse anatomical sites, demanding tailored formulations to precisely regulate gel curing rates and drug content. Furthermore, the exudate from the wound emerges as a significant variable affecting drug release, which proves challenging to accurately control through in vitro experimentation.

## Figures and Tables

**Figure 1 pharmaceutics-16-00875-f001:**
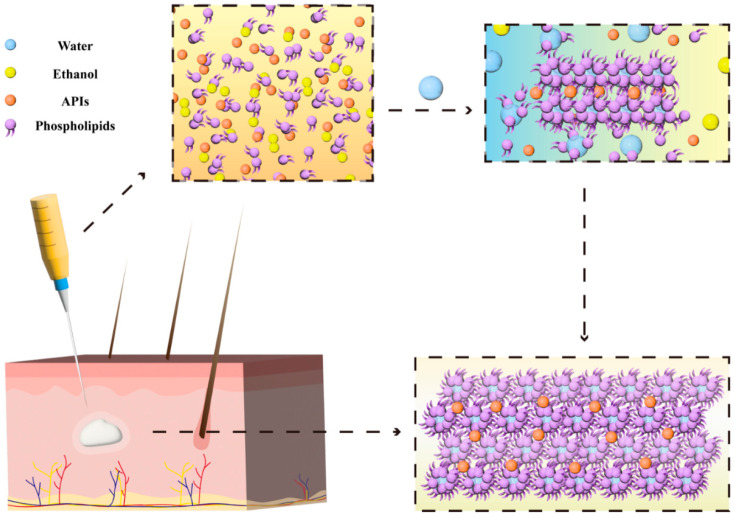
Mechanism of PPSG formation.

**Figure 2 pharmaceutics-16-00875-f002:**
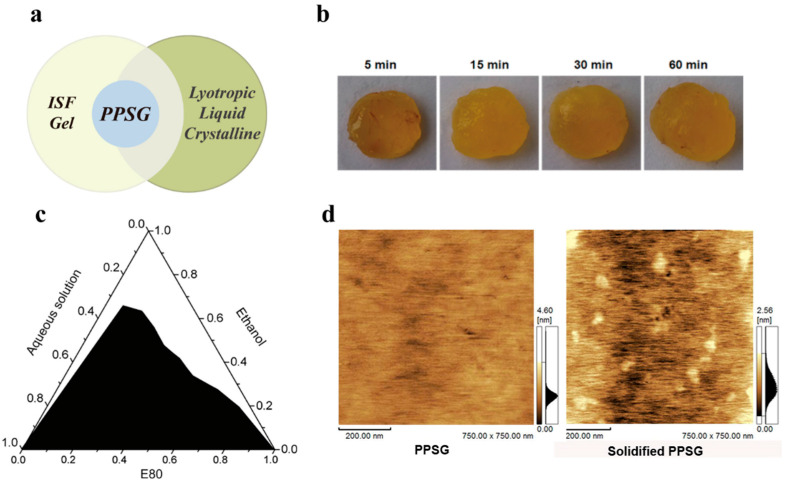
(**a**) Category of PPSG. (**b**) Photos of PPSG after injection. (**c**) Ternary phase diagram of PPSG. (**d**) AFM graphs of PPSG before and after the phase transition. Reprinted with permission from Elsevier, 2015 [25].

**Figure 3 pharmaceutics-16-00875-f003:**
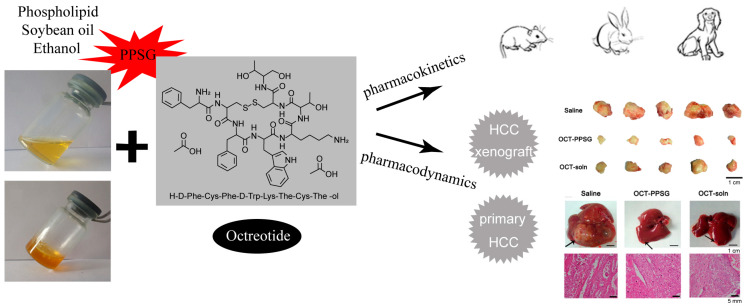
OCT–PPSG showed desirable pharmacokinetic profiles in rats, rabbits, and Beagle dogs with antitumor activity in primary and xenograft models of hepatocellular carcinoma (HCC). A heavy enlargement and several grayish white nodules and foci were observed on the peripheral surface of the liver as the features of HCC of the saline group, while disappeared in the OCT-PPSG and the OCT-soln group pointed by the black arrow. Reprinted with permission from Elsevier, 2019 [26].

**Figure 4 pharmaceutics-16-00875-f004:**
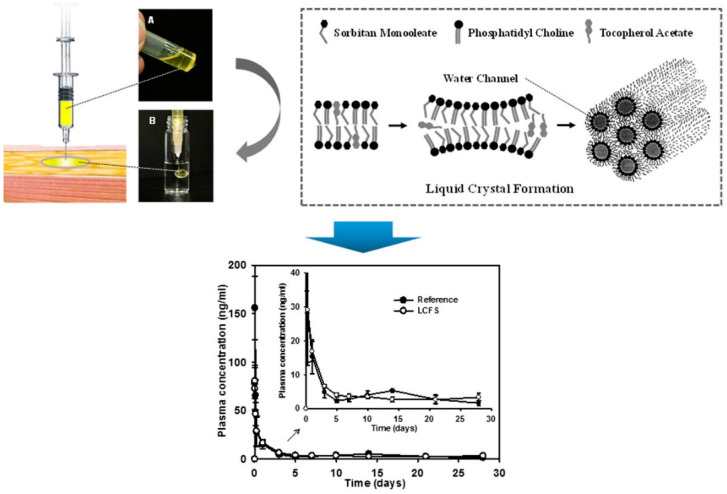
A new injectable liquid crystal system for the one-month delivery of leuprolide. Images of LCFS containing leuprolide acetate before injection (**A**) and liquid crystalline mesophase formed in PBS (pH 7.4) after injection (**B**) are shown. Reprinted with permission from Elsevier, 2014 [22].

**Figure 5 pharmaceutics-16-00875-f005:**
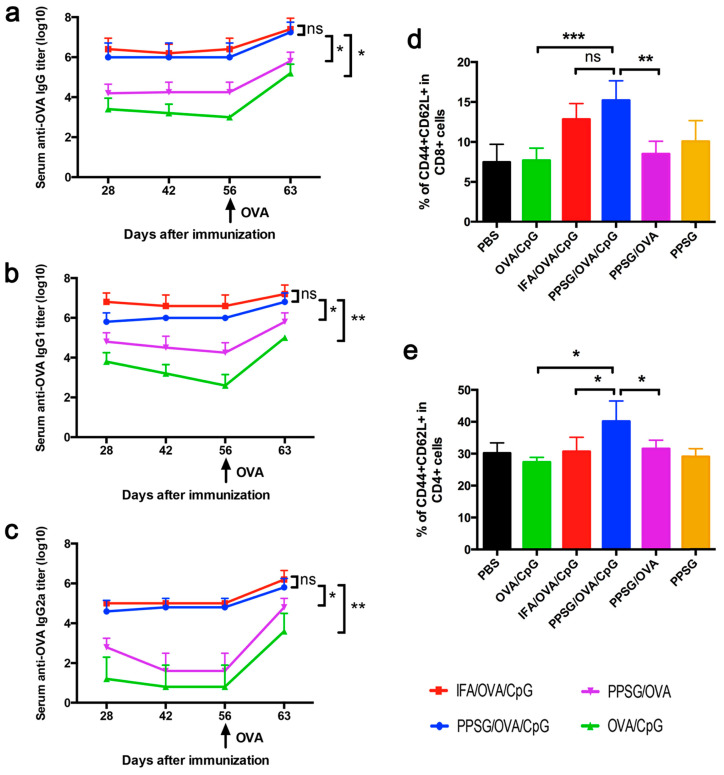
OVA–PPSG induces strong and persistent immune responses in mice. The serum titers of OVA-specific IgG, IgG1 and IgG2a were measured at day 28, 42, 56 and 63 after the first immunization and the results are presented in Figure (**a**–**c**) respectively. The percentages of CD44^+^CD62L^+^ cells among all CD8^+^T cells and of CD44^+^CD62L^+^cells among all CD4^+^T cells determined by flow cytometry are presented in Figure (**d**,**e**) respectively. Data are shown as mean ± SD (n = 3~5). Reprinted with permission from Elsevier, 2016 [28]. * *p* < 0.05, ** *p* < 0.01, *** *p* < 0.001, ns: not significant.

**Figure 6 pharmaceutics-16-00875-f006:**
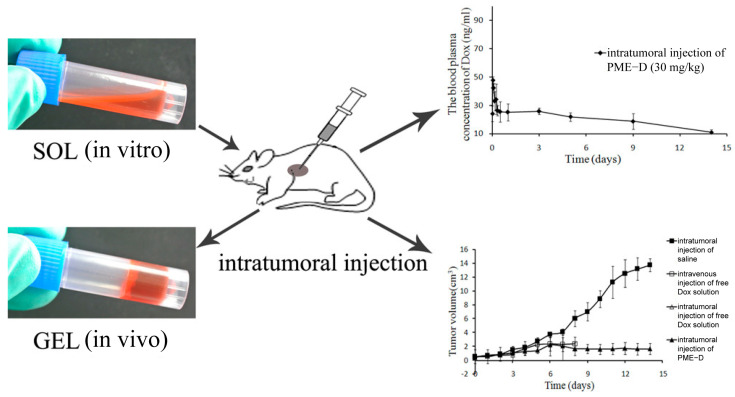
A novel doxorubicin-loaded ISF gel, with a high concentration of phospholipids, for intratumoral drug delivery. This system was in a solid state with low viscosity in vitro and turned into a solid or semisolid gel in situ after injection. When loaded with doxorubicin (Dox), this system exhibited sustained release without burst release for more than 14 days and, notably, antitumor efficiency in S180 sarcoma tumor-bearing mice after a single intratumoral injection. Reprinted with permission from the American Chemical Society, 2014 [16].

**Figure 7 pharmaceutics-16-00875-f007:**
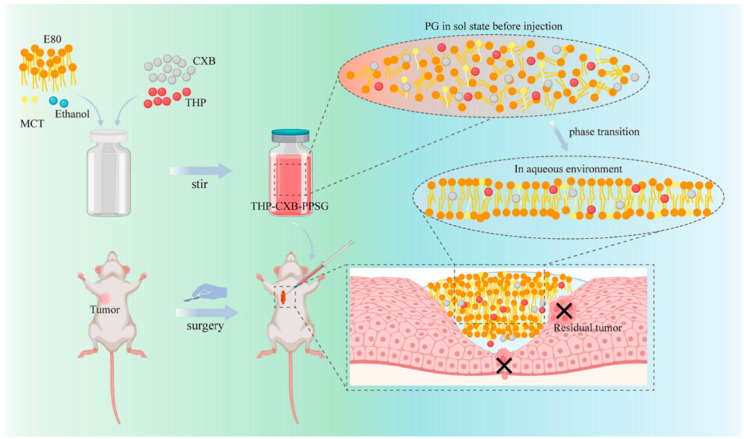
An ISF gel co-loaded with pirarubicin and celecoxib inhibits postoperative recurrence and metastasis of breast cancer. Reprinted with permission from Elsevier, 2014 [81].

**Figure 8 pharmaceutics-16-00875-f008:**
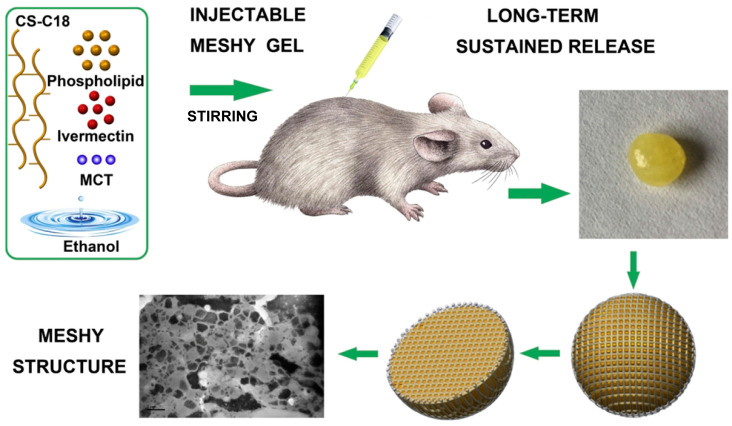
Schematic illustration of the organization of meshy gel system, gelatinized by ethanol diffusion and precipitation of high concentrations of phospholipids and CS-C18 to form an in situ drug depot with a meshy structure. Abbreviations: CS-C18, octadecylamine-modified chondroitin sulfate; MCT, medium chain triglyceride. Reprinted with permission from Elsevier, 2019 [38].

**Table 1 pharmaceutics-16-00875-t001:** Summary of PPSG systems and their application.

		Encapsulated	Application	Year	References
Peptides		Exenatide (synthetic exendin-4, EXT)	Type 2 Diabetes	2016	[21]
		Leuprolide Acetate (LA)	Prostate Cancer	2014	[22]
		Leuprolide Acetate (LA)	Prostate Cancer	2016	[23]
		Thymopentin (TP5)	Autoimmune Disease	2019	[24]
		Octreotide Acetate (OCT)	Watery Diarrhea	2015	[25]
		Octreotide Acetate (OCT)	Watery Diarrhea	2016	[26]
		Octreotide Acetate (OCT)	Watery Diarrhea	2020	[27]
Proteins		Ovalbumin (OVA)	Vaccine Adjuvant	2016	[28]
		Insulin	Diabetes	2019	[29]
Chemotherapeutics	Tumor	Doxorubicin (Dox)	S180 Sarcoma Tumors	2014	[16]
		Paclitaxel (PTX)	Glioma	2017	[30]
		5-Fluotouracil (5FU) and Magnesium Oxide (MgO)	Breast Cancer	2018	[31]
	Arthritis	Methotrexate and Dexamethasone	Rheumatoid Arthritis	2022	[32]
		Celastrol	Rheumatoid Arthritis	2023	[33]
	Anesthesia	Ropivacaine (RO)	Local Anesthetic	2017	[34]
		Brexpiprazole	Schizophrenia	2023	[35]
		Ziprasidone	Schizophrenia	2020	[36]
	Others	2,4-dinitrophenol (DNP)	Nonalcoholic Fatty Liver Disease	2017	[37]
		Ivermectin (IVM)	Malaria	2019	[38]
		Progesterone	Threatened Abortion	2022	[39]
		Alendronate Sodium Nano Emulsion	Osteoporosis	2023	[40]
		Dabigatran etexilate (DABE)	Venous Thrombosis	2017	[41]

**Table 2 pharmaceutics-16-00875-t002:** Excipients used in PPSGs.

Phospholipids (PL)	Solvents	Oils
		Soybean oil
	Ethanol	Sorbitan monooleate
**Phosphatidyl choline**		(SMO; Span-80)
(PC)		Medium chain oil
		(MCT)
	NMP	Glycerol monooleate
		(GMO)
**Phosphatidyl ethanolamine**		Glycerol dioleate
(PE)		(GDO)
	Isopropanol	Glycerol trioleate
		(GTO)

**Table 3 pharmaceutics-16-00875-t003:** PPSG-based peptide/protein delivery systems.

	Encapsulated	Sequence	Molecular Weight (Mw)	Application	Excipients	Release Time in PPSG	Year	References
Peptide	Octreotide (OCT)	8AA	1019.24	Hepatocellular Carcinoma	E80/85% ethanol/soybean oil (70:15:15, *w/w/w*)	One month	2016	[26]
	Exenatide (synthetic exendin-4, EXT)	39AA	4186.66	Type 2 Diabetes	S100/90% ethanol/MCT (70:15:15, *w/w/w*)	35 days	2016	[21]
	Thymopentin (TP5)	5AA	679.77	Autoimmune Disease	E80/ethanol/medium chain triglyceride (MCT) (70:15:15, *w/w/w*)	One month	2019	[24]
	Leuprolide Acetate (LA)	9AA	1269.4502	Prostate Cancer	S100:ethanol:MCT (70:15:15, *w/w/w*)	15 days	2016	[23]
	Leuprolide Acetate (LA)	9AA	1269.4502	Prostate Cancer	SMO, phosphatidyl choline, tocopherol acetate, Tween 80, and ethanol (33:45:10:2:10, *w/w*%)	30 days	2014	[22]
Protein	Ovalbumin	386AA	45KD	Vaccine Adjuvant	E80/ethanol/soybean oil (70:15:15, *w/w/w*)	One month	2016	[28]
	Insulin	51AA	5700	Diabetes	E80/ethanol–citric acid buffer (pH 3.0, 85/15)/MCT (70:15:15, *w/w/w*)	8 days	2019	[29]

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
