# Peer review of "The Advances in Phospholipids-Based Phase Separation Gels for the Sustained Release of Peptides, Proteins, and Chemotherapeutics"

_pharmaceutics, 2024, doi:10.3390/pharmaceutics16070875_

Round 1
Reviewer 1 Report
Comments and Suggestions for Authors
The review is devoted to phospholipids-based phase separation gels for sustained release of peptides, proteins, and chemotherapeutics. It is structured relatively well, but the style of writing is a little bit careless.
I have the following suggestions for improvement:
1. Page 3, Rewrite the sentence: "In the past few decades, a series of novel phospholipid-based phase separation gels (PPSGs) have been developed and explored for the sustained release of peptides, proteins, and chemotherapeutics.()"
2. Clarification needed in the introduction on sol-to-gel and solution-to-gel transitions: The sol-to-gel transition involves the formation of a gel-like material from a colloidal suspension (sol) of solid particles or aggregates dispersed in a liquid. In contrast, the solution-to-gel transition occurs when a homogeneous solution undergoes a phase change to form a gel-like material, where the dissolved molecules or polymers self-assemble or cross-link to create a three-dimensional network that traps the solvent.
3. Distinction needed between oils, lipids, and phospholipids: Oils are a type of lipid, which are a class of organic compounds that are generally insoluble in water but soluble in nonpolar solvents. Phospholipids are a specific type of lipid that are the primary structural components of cell membranes, with a hydrophilic head and two hydrophobic fatty acid tails. GMO is a lipid!
4. Clarification on the mechanism depicted in Figure 1: The mechanism shown in Figure 1 appears to be a solution-to-gel transition, where the dissolved phospholipid molecules self-assemble into a three-dimensional network to form the gel-like structure.
5. The figure number on page 6 should be Figure 3, not Figure 1.
6. Rewrite the sentence: "However, several barriers appear during the development of protein and peptide-drugs [26,49,50]: 1) with their digestible nature, administration is limited to injections, which requires much higher compliance from the patients; 2) due to their relatively complicated chemical structure, stability varies largely under different pH environments and the appearance of endogenous enzymes; 3) shorter half-life demands higher frequency of dosing."
7. Correction of "Error! Reference source not found" needed: The instances of "Error! Reference source not found" should be replaced with the appropriate references or figures.
8. Inclusion of structural data: The review should include relevant structural data, such as electron microscopy, X-ray scattering, or diffraction, to provide a more comprehensive structural understanding of the phospholipid-based phase separation gel systems.
9. The figure number on page 13 should be Figure 5, not Figure 2.
10. The figure number on page 13 should be Figure 6, not Figure 3.
11. The figure number on page 14 should be Figure 7, not Figure 4.
12. The authors may find helpful a recent paper on the topic of the review that investigates similar lipid based drug delivery systems: Composition-Switchable Liquid Crystalline Nanostructures as Green Formulations of Curcumin and Fish Oil, ACS Sustainable Chem. Eng. 2021, 9, 44, 14821–14835. https://doi.org/10.1021/acssuschemeng.1c04706
Comments on the Quality of English Language
The revision at the paragraph level could improve the clarity and cohesion of the English language usage. However, the typing itself does not appear to have any notable errors.
Author Response
The review is devoted to phospholipids-based phase separation gels for sustained release of peptides, proteins, and chemotherapeutics. It is structured relatively well, but the style of writing is a little bit careless.
Response: We appreciate the Reviewer’s summary of our study and positive comments. Please see the specific response to each comment below.
Comment 1 Page 3, Rewrite the sentence: “In the past few decades, a series of novel phospholipid-based phase separation gels (PPSGs) have been developed and explored for the sustained release of peptides, proteins, and chemotherapeutics.()”
Response: We appreciate the Reviewer’s suggestion and this comment has been addressed.
Action: “In the last few decades, researchers have developed a novel category of drug delivery systems, PPSGs. for the controlled release of peptides, proteins, and chemotherapeutics (Table 1).” (Page 3, line 90-92, in the revised manuscript)
Comment 2 Clarification needed in the introduction on sol-to-gel and solution-to-gel transitions: The sol-to-gel transition involves the formation of a gel-like material from a colloidal suspension (sol) of solid particles or aggregates dispersed in a liquid. In contrast, the solution-to-gel transition occurs when a homogeneous solution undergoes a phase change to form a gel-like material, where the dissolved molecules or polymers self-assemble or cross-link to create a three-dimensional network that traps the solvent.
Response: We appreciate the Reviewer’s suggestion and this comment has been addressed.
Action: “sol-to-gel” is revised to “solution-to-gel transition”. (Page 3, line 95, in the revised manuscript)
Comment 3 Distinction needed between oils, lipids, and phospholipids: Oils are a type of lipid, which are a class of organic compounds that are generally insoluble in water but soluble in nonpolar solvents. Phospholipids are a specific type of lipid that are the primary structural components of cell membranes, with a hydrophilic head and two hydrophobic fatty acid tails. GMO is a lipid!
Response: Thanks for your advice. Lipids in this manuscript include oils, phospholipids and GMO.
Comment 4 Clarification on the mechanism depicted in Figure 1: The mechanism shown in Figure 1 appears to be a solution-to-gel transition, where the dissolved phospholipid molecules self-assemble into a three-dimensional network to form the gel-like structure.
Response: Thanks for your suggestion and we have revised it.
Action: “Mechanistically, the solution-to-gel transition exhibits several benefits in comparison to other gel formation mechanisms [16]. The dissolved phospho-lipid molecules self-assemble into a three-dimensional network to form the gel-like structure.” (Page 3, line 95-97, in the revised manuscript)
Comment 5 The figure number on page 6 should be Figure 3, not Figure 1.
Response: Per the Reviewer’s comment, we have modified the figure number on page 6 in the revised manuscript.
Action: Change Figure 1 to Figure 3. (Page 6, line 179, in the revised manuscript)
Comment 6 Rewrite the sentence: "However, several barriers appear during the development of protein and peptide-drugs [26,49,50]: 1) with their digestible nature, administration is limited to injections, which requires much higher compliance from the patients; 2) due to their relatively complicated chemical structure, stability varies largely under different pH environments and the appearance of endogenous enzymes; 3) shorter half-life demands higher frequency of dosing."
Response: We appreciate the Reviewer’s suggestion and this comment has been addressed.
Action: “However, there are several challenges in the development of protein and peptide drugs [26,49,50]: 1) Administration is limited to injections due to their digestible nature, requiring higher patient compliance; 2) Stability varies largely under different pH environments and the appearance of endogenous depending on their chemical structure; 3) Dosing frequency is high because of their relatively short half-life.” (Page 6, line 198-203, in the revised manuscript)
Comment 7 Correction of "Error! Reference source not found" needed: The instances of "Error! Reference source not found" should be replaced with the appropriate references or figures.
Response: Per the Reviewer’s comment, we have corrected "Error! Reference source not found".
Action: Delete all of “Error! Reference source not found” and replace them with figures.
Comment 8 Inclusion of structural data: The review should include relevant structural data, such as electron microscopy, X-ray scattering, or diffraction, to provide a more comprehensive structural understanding of the phospholipid-based phase separation gel systems.
Response: Per the Reviewer’s comment, we have added the atomic force microscopy (AFM) graphs of PPSG in the revised manuscript.
Action: We added AFM graphs of PPSG before and after phase transition in Figure 2d. (Page 5, line 158, in the revised manuscript)
Comment 9-11 The figure number on page 13 should be Figure 5, not Figure 2. The figure number on page 13 should be Figure 6, not Figure 3. The figure number on page 14 should be Figure 7, not Figure 4.
Response: Per the Reviewer’s comment, we have modified the figure number on pages 13-14 in the revised manuscript.
Action: Change Figure 2 to Figure 5. (Page 13, line 366, in the revised manuscript); Change Figure 3 to Figure 6 (Page 13, line 389, in the revised manuscript); Change Figure 4 to Figure 7 (Page 14, line 405, in the revised manuscript); Change Figure 5 to Figure 8 (Page 14, line 439, in the revised manuscript).
Comment 12 The authors may find helpful a recent paper on the topic of the review that investigates similar lipid based drug delivery systems: Composition-Switchable Liquid Crystalline Nanostructures as Green Formulations of Curcumin and Fish Oil, ACS Sustainable Chem. Eng. 2021, 9, 44, 14821–14835.
Response: Thank you for providing the helpful literature. We have cited it in the revised manuscript.
Action: Add “For example, synchrotron small-angle X-ray diffuser and transmission electron microscope could be used to analyze the nanostructures [98].” (Page 16, line 497-499, in the revised manuscript)

Reviewer 2 Report
Comments and Suggestions for Authors
Dong et al. review on in situ forming gel systems based on phase separation of phsopholipids. Initially in a liquid form, the systems undergo solvent (water-ethanol) exchange upon administration and form solid or semi-solid gels at the injection site (implants) that can serve as platforms, “reservoirs”, depots or carriers for peptides, proteins, and chemotherapeutics. The systems exhibit abilities to extend the release over weeks and months thus allowing reduction of dosing frequency. The review is well-written, comprehensive, and systematic and contains description of various systems as well as challenges and perspective. My recommendations and suggestions for improvements concern mostly the technical issue and are listed below.
- Introduce abbreviations in the first mentioning in the text and only once. For example, “API” is introduced at the end of the manuscript (page 15) but it is repeatedly used earlier in the text.
- Opposite to the above, some abbreviations are multiple introduced (e.g., PPSG, ISF).
- There are two Tables 1 (page ¾ and page 8). Neither of them is mentioned, rather than discussed.
- Figure 2a and 2c are neither mentioned, nor discussed. Instead, the discussion of Figure 2 starts with Figure 2d.
- Carefully check the figure numbering. There are two Figures 1, two Figures 2, two Figures 4 as well as Figure 3, that supposedly correspond to Figure 7.
- Avoid phrases like “to develop the development”
- The number of section 3.3.2. is not consistent.
- Delete “1.3.2 Delivery of Exenatide (EXT).” from the figure caption of Figure 1, page 6.
Comments on the Quality of English LanguageMinor editing of English language required
Author Response
Reviewer #: 2
Recommendation: Dong et al. review on in situ forming gel systems based on phase separation of phsopholipids. Initially in a liquid form, the systems undergo solvent (water-ethanol) exchange upon administration and form solid or semi-solid gels at the injection site (implants) that can serve as platforms, “reservoirs”, depots or carriers for peptides, proteins, and chemotherapeutics. The systems exhibit abilities to extend the release over weeks and months thus allowing reduction of dosing frequency. The review is well-written, comprehensive, and systematic and contains description of various systems as well as challenges and perspective.
Response: We appreciate the Reviewer’s summary of our study and positive comments. Please see the specific response to each comment below.
Comment 1-2 Introduce abbreviations in the first mentioning in the text and only once. For example, “API” is introduced at the end of the manuscript (page 15) but it is repeatedly used earlier in the text. Opposite to the above, some abbreviations are multiple introduced (e.g., PPSG, ISF).
Response: Per the Reviewer’s comment, we have introduced the abbreviations only once in the revised manuscript.
Comment 3 There are two Tables 1 (page 3/4 and page 8). Neither of them is mentioned, rather than discussed.
Response: Per the Reviewer’s comment, we have renumbered the tables.
Action: We have renumbered the tables.
Comment 4 Figure 2a and 2c are neither mentioned, nor discussed. Instead, the discussion of Figure 2 starts with Figure 2d.
Response: According to the Reviewer’s comment, we have mentioned Figure 2a and 2 in the revised manuscript. The discussion of Figure 2 also starts with Figure 2a.
Action: Rewrite paragraph “PPSG is a special ISF gel (Figure 2a), which composed of three basic components: phospholipids, low-percentage ethanol, and pharmaceutical oil. The phospholipids are the major components of cell membranes and serve as a drug depot in PPSG. The ethanol works as a solvent for phospholipids to increase spreadability (reduce viscosity) and therefore ensure the smooth injection. When this system is in contact with an aqueous environment, the solvent exchange occurs immediately at the interface, consequently inducing solidification of PPSG (Figure 2b). A layer of shell structure quickly forms at the surface of the gel. Then internal ethanol dissipates out of the system, while water ingresses gradually via diffusion. This exchange of solvents results in the increase of viscosity and solution-to-gel transition, which turns phospholipids into a drug depot. (Figure 1). The ternary phase diagram of PPSG shows a big phase transition area (Figure 2c), which also illustrates that PPSG can easily transit to gel in an aqueous environment. Furthermore, when PPSG is observed using atomic force microscopy (AFM), there is no bright spots in the AFM graph, indicating that the phospholipid in PPSG is uniformly dissolved and well-distributed. After PPSG transforming into gel in contact with an aqueous environment, its AFM graph shows many bright spots, which illustrates the presence of solidified phospholipid particles with a certain size. This result suggests that the transformation of PPSG into gel leads to the precipitate and solidification of phospholipid, consequently becoming a reservoir for sustained-release of drugs (Figure 2d).” (Page 4, line 111-129, in the revised manuscript)
Comment 5 Carefully check the figure numbering. There are two Figures 1, two Figures 2, two Figures 4 as well as Figure 3, that supposedly correspond to Figure 7.
Response: Per the Reviewer’s comment, we have modified the figure number in the revised manuscript.
Action: Change Figure 2 to Figure 5. (Page 13, line 366, in the revised manuscript); Change Figure 3 to Figure 6 (Page 13, line 389, in the revised manuscript); Change Figure 4 to Figure 7 (Page 14, line 405, in the revised manuscript); Change Figure 5 to Figure 8 (Page 14, line 439, in the revised manuscript).
Comment 6 Avoid phrases like“to develop the development”
Response: Per the Reviewer’s comment, we have modified the related phrases.
Action: Delete “the development of”. (Page 6, line 197, in the revised manuscript)
Comment 7 The number of section 3.3.2. is not consistent.
Response: Per the Reviewer’s comment, we have modified the section number in the revised manuscript.
Action: Change “3.3.2” to “4.2”. (Page 9, line 287, in the revised manuscript)
Comment 8 Delete “1.3.2 Delivery of Exenatide (EXT).” from the figure caption of Figure 1, page 6.
Response: Per the Reviewer’s comment, we have deleted“1.3.2 Delivery of Exenatide (EXT).” in the revised manuscript.

Round 2
Reviewer 1 Report
Comments and Suggestions for Authors
The authors corrected and improved the paper. I do not have other suggestions.